# A Nano-Emulsion Containing Ceramide-like Lipo-Amino Acid Cholesteryl Derivatives Improves Skin Symptoms in Patients with Atopic Dermatitis by Ameliorating the Water-Holding Function

**DOI:** 10.3390/ijms232113362

**Published:** 2022-11-01

**Authors:** Mariko Takada, Yuko Ishikawa, Kayoko Numano, Shinichi Hirano, Genji Imokawa

**Affiliations:** 1Center for Bioscience Research & Education, Utsunomiya University, 350 Mine, Utsunomiya 321-8505, Tochigi, Japan; 2ToutVert Co., Ltd., 2-2-1, Senbanishi, Minoh 562-0036, Osaka, Japan; 3Quines Square Dermatological Clinic, 8th Fl. Queen’s Tower C, 2-3-5, Minatomirai, Nishi-Ku, Yokohama 220-0012, Kanagawa, Japan

**Keywords:** atopic dermatitis, skin symptoms, water content, barrier function, lamellae structure

## Abstract

Because ceramide-like lipo-amino acid cholesteryl derivatives can exert a bound water-holding function due to their lamellae-forming properties, in this study, we determined if topical application of those derivatives to atopic dry skin would elicit an ameliorative effect on skin symptoms, at least on its water-holding function. In this clinical study, daily treatment with a nano-emulsion containing 10% phytosteryl/octyldodecyl lauroyl glutamate (POLG) significantly (*p* < 0.0001) improved skin symptoms, including dryness/scaling, itchiness and stimulus sensations, in the non-lesional skin of patients with atopic dermatitis (AD) at 3 and at 6 weeks compared with week 0. Those significant improvements in skin symptoms were accompanied by a significantly enhanced water content (conductance) and a significant improvement of roughness (SESC) and smoothness (SESM) values measured using a Visioscan at 3 and 6 weeks. Those effects appeared concomitant with a significantly increased corneocyte size, a significantly down-regulated degree of thick abrasions, and a significant impairment of the corneocyte lipid envelope at 6 weeks. Thus, our clinical study suggests, for the first time, that topical application of the POLG nano-emulsion has the distinct potential to ameliorate atopic dry skin symptoms, particularly scaling and itchiness, in the skin of patients with AD. Those effects result from alleviation of the disrupted water-holding function probably due to the increased supply of lamellae structures into the stratum corneum despite the failure to improve barrier function.

## 1. Introduction

Atopic dermatitis (AD) is a recurrent type of skin dermatitis that has a high susceptibility to itching, irritants, and allergens, even in non-lesional skin, called atopic dry skin, with severe dry appearances (dryness) [1]. AD is characterized clinically by severe dry skin, and, functionally, by cutaneous barrier disruption and impaired water-holding function, properties that reside in the stratum corneum (SC) [1,2]. These functional abnormalities of the SC are mainly attributable to significantly decreased levels of total ceramides [1,3,4,5,6,7] and altered ceramide profiles in the SC [8]. However, the latter is not AD-specific, being attributable to a facet of previously evoked cutaneous inflammation [8]. As an etiologic and essential biofactor that can provoke a ceramide deficiency even under the normal ceramide metabolism observed in the AD skin [9,10], we have discovered a novel enzyme, sphingomyelin (SM) deacylase, which cleaves the N-acyl linkage of SM and glucosylceramide [4,9,11,12,13] by competing with acid sphingomyelinase and β-glucocerebrosidase to yield their lysoforms sphigosylphosphorylcholine (SPC) and glucosylsphingosin (GS), respectively, instead of ceramide, resulting in the ceramide deficiency in AD skin. SM deacylase activity is significantly up-regulated in the AD skin, including the lesional epidermis, as well as the involved and uninvolved SC, but not in the skin of patients with contact dermatitis or chronic eczema, compared with healthy controls (HCs) [4,11,12]. Those reaction products (SPC and GS) occur to a greater extent in the involved and uninvolved SC of AD skin compared with chronic eczema or contact dermatitis, as well as the healthy control, with significant correlations of the increased levels of SPC and GS with the decreased levels of total ceramides or acylceramide [4,6]. Using rat skin, we have succeeded to purify SM deacylase to homogeneity (a single spot by 2D-SDS-PAGE) to identify it as the β-subunit of acid ceramidase (aCDase), an enzyme consisting of α- and β-subunits linked by amino-bonds and a single S-S bond [14].

The combination of the disrupted barrier and water-holding functions in atopic dry skin is closely linked to the disease severity of AD [15,16], which suggests that the barrier abnormality, as well as the water deficiency, are elicited as a facet of the previously provoked dermatitis. Thus, the atopic dry skin that still exists even after amelioration of skin inflammation, as recognized by erythema, papules, and abrasion, usually by corticosteroid treatments, plays a refractory role in triggering the recurrence of dermatitis. Based on such clinical atopic dry skin features, owing to a failure to develop a specific inhibitor of SM deacylase, a skin care system that can ameliorate the functional abnormalities of the SC is essential to prevent the recurrence of AD.

The most reasonable approach to improving these abnormalities in the SC is to topically apply natural skin ceramides at a high concentration to compensate for the ceramide deficiency in the SC of AD skin. Although we have already confirmed the distinct efficacy of natural ceramides by topical application on dry skin caused by treatments with solvents or surfactants [17,18,19,20], there is limited availability of natural ceramides due to their cost and the difficulty of incorporating them into creams or lotions at concentrations high enough to exert ameliorating effects on the functional abnormalities of the SC in AD skin. To resolve this problem, we have examined the physico-chemical properties of natural ceramides that are essential to exert water-holding properties and barrier functions. We found that the potential to form lamellar structures comprising multiple complexes of lipid components and water molecules is closely associated with the water-holding capacity [17,21,22], whereas additional structural characteristics of those lamellar structures are required for the full achievement of barrier function [23,24]. In this line of analysis of physico-chemical properties, we developed a synthetic pseudoceramide (pCer), N-(3-hexadecyloxy-2-hydroxypropyl)-N-2-hydroxyethylhexadedanaimde, which has the capacity to form multi-concentric lamellar vesicles in aqueous solution, and to form multi-lamellar structures between the SC layers. Those were accompanied by the incorporation of water into the lamellar structures as non-evaporable bound water, the formation of which is essential for the water-holding properties of the SC even under extremely dry conditions [17,21,22,25]. pCer also has a limited availability because of its cost and the difficulty of incorporating it into emulsions such as creams or lotions, as occurs for natural ceramides. 

We have already reported that the atopic dry skin and water deficiency that occurs in the nonlesional skin of patients with AD [8], or that is induced by treatment of normal healthy skin with surfactants or solvents [17,21,25], can be ameliorated or mitigated by the topical application of pCer at relatively high concentrations (>3.0%). That in vivo efficacy strongly supports our hypothesis that the ability of chemicals to form multi-lamellar structures is essentially ascribed to the water-holding property at least [17], and possibly to the barrier function [23], the amelioration of which leads to an improvement of the dry skin in AD patients [24,26,27,28]. In our search for various chemicals that have the ability to form multi-lamellar structures, we have identified lipo-amino acid cholesteryl derivatives [29,30]. Phytosteryl/octyldodecyl lauroyl glutamate (POLG), one of those lipo-amino acid cholesteryl derivatives, was reported to have a structural resemblance to ceramide, and to have a distinct potential to form multi-lamellar vesicles, accompanied by recovery effects on disrupted water-holding properties induced by surfactant or solvent treatments [29,30]. Because POLG has a potential for use in terms of its reasonable cost and high availability in emulsions, in this study, we determined if the topical application of a 10% POLG containing emulsion to atopic dry skin would elicit an ameliorative effect on atopic dry skin symptoms, at least on the water-holding function. 

## 2. Results

### 2.1. Effect of the POLG Nano-Emulsion on Atopic Dry Skin during the 6 Weeks of Treatment 

Treatment with the POLG nano-emulsion induced a slight decrease in dryness/scaling scores at 3 weeks, and induced a significant decrease (*p* < 0.01) in those scores at 6 weeks (Figure 1), whereas the same treatment did not cause any changes in erythema (redness), papules, or abrasions at 3 or at 6 weeks (Figure 2, Figure 3 and Figure 4). Stimulus sensations were significantly ameliorated after treatment with the POLG nano-emulsion for 3 weeks (*p* < 0.001) and for 6 weeks (*p* < 0.0001) compared with week 0 (Figure 5). Itchiness was also significantly reduced after treatment with the POLG nano-emulsion for 3 weeks (*p* < 0.05) and for 6 weeks (*p* < 0.0001) compared with week 0 (Figure 6).

### 2.2. Clinical Improvement by the POLG Nano-Emulsion

The POLG nano-emulsion showed 12.0, 8.0, 72.0, and 8.0% distribution after 3 weeks of treatment, and 12.0, 40.4, 44.0, and 4.0% distribution after 6 weeks of treatment with marked improvement, moderate improvement, slight improvement, or no improvement, respectively (Figure 7). Those improvement degrees were significantly different (*p* < 0.0001) both at 3 and at 6 weeks compared with week 0 (Figure 7), which suggests that there was a distinct efficacy for atopic dry skin symptoms.

### 2.3. Skin Surface Evaluation by Visioscan 

Skin surface evaluation by Visioscan revealed that 6 weeks of treatment with the POLG nano-emulsion induced significant reductions in the parameters for scaliness (SESC) and smoothness (SESM) (Figure 8A,B), although the same 6 weeks of treatment did not elicit any significant change in the parameter for roughness (SER) (Figure 8C). These findings indicate that 6 weeks of treatment with the POLG nano-emulsion significantly mitigated scaling at the skin surface (Figure 8D) to facilitate skin smoothness compared to the onset of treatment.

### 2.4. Effect of the POLG Nano-Emulsion on TEWL and Conductance Values during the 3 and 6 Weeks of Treatment 

Treatment with the POLG nano-emulsion induced a significant amelioration in conductance values, with significant and marked increases at 3 and at 6 weeks (33.24, 57.67, 66.37/173.5%, and 199.7%, respectively) compared with week 0 (Figure 9A). On the other hand, the treatment with the POLG nano-emulsion elicited a slight, but significant, deterioration in trans-epidermal water loss (TEWL) values, with very slight, but significant, increases at 3 and at 6 weeks (8.03, 9.53, 8.96/118.7%, and 111.6%, respectively) compared with week 0 (Figure 9B). However, a significantly increased TEWL value of 2.0 g/m^2^ h is substantially meaningless based on the evidence that less than 10 g/m^2^ h TEWL generally occurs in the skin of healthy subjects. These findings suggest that the POLG nano-emulsion has a considerable potential to increase the water content in the SC, whereas it has no significant ability to strengthen the barrier function, although the AD patients enrolled in this study had no deficiency in their barrier function compared with healthy controls [15]. 

### 2.5. Corneocyte Size, Degree of Thick Abrasions, and Nile Red Dye Staining

The 6 weeks of treatment with the POLG nano-emulsion induced a significant increase in corneocyte size (Figure 10A), which suggests that there was a down-regulated level of turnover rate elicited by the POLG nano-emulsion. The degree of thick abrasions was significantly decreased by treatment with the POLG nano-emulsion, probably reflecting an amelioration of the desquamation (Figure 10B). The treatment with the POLG nano-emulsion also elicited a significant increase in Nile red dye staining (Figure 10C), which suggests that the disrupted feature of the lipid envelope is mitigated by that treatment. 

## 3. Discussion

In this clinical study, the daily continuous use of the 10% POLG nano-emulsion significantly (*p* < 0.001) improved several dry skin symptoms, including dryness/scaling, itchiness, and stimulus sensations in the non-lesional skin of patients with AD at 3 and at 6 weeks compared with week 0. Based on general clinical evaluation, the POLG nano-emulsion had 12.0, 8.0, 72.0, and 8.0% distribution after 3 weeks of treatment, and 12.0, 40.4, 44.0, and 4.0% distribution after 6 weeks of treatment, with marked improvement, moderate improvement, slight improvement, and no improvement, respectively. As we have already performed similar clinical studies [24,26,27,28] on the forearm skin of AD patients using pseudo-ceramide, urea, and mucopolysaccharide-containing creams or lotions, those comparisons for each type of clinical improvement allow us to understand the relative level of clinical efficacy of the POLG nano-emulsion as follows: (1) our previous comparative clinical study using CER and HIRU creams [26] demonstrated that an 8% synthetic pCer, N-(3-hexadecyloxy-2-hydroxypropyl)-N-2-hydroxyethylhexadedanaimde-containing (CER) cream(W/O type) had a 50.0, 35.7, 14.3, and 0% distribution after 4 weeks of treatment, with marked improvement, moderate improvement, slight improvement, and no improvement, respectively. The Hirudoido^®^(HIRU) cream (W/O type) had 0, 15.4, 80.8, and 3.9% distribution after 4 weeks of treatment. (2) In another comparative clinical study using the CER and HIRU creams [28], the CER cream had 15.2, 68.7, 3.7, and 12.4% distribution after 6 weeks of treatment, with marked improvement, moderate improvement, slight improvement, and no improvement, respectively. The HIRU cream had 18.6, 31.3, 37.5, and 12.6% distribution after 6 weeks of treatment. (3) In our similar comparative study [31] using the CER and 20% urea-containing creams (urea cream), the CER cream had 12.0, 48.0, 40.3, and 0% distribution after 6 weeks of treatment, with marked improvement, moderate improvement, slight improvement, and no improvement, respectively. The urea cream had 0, 10.0, 20.0, and 70.0% distribution after 6 weeks of treatment. These comparative evaluations for the efficacy of each treatment indicate that the POLG nano-emulsion is situated at a middle level of clinical efficacy, being in the order of: CER cream > POLG nano-emulsion > HIRU cream > urea cream, and is clinically more effective in ameliorating atopic dry skin than the HIRU cream, which is currently being used as an anti-atopic medicine in Japan. 

In this study, improvements of atopic dry skin symptoms were confirmed by a dermatologist’s (KN) clinical scoring and by instrumental measures for water-holding functions (conductance), although improvements by instrumental measures for barrier functions (TEWL) were not detectable. It is likely that the lack of improvement in the barrier function may be attributable either to the healthy control levels of TEWL values (<10 g/m^2^/h), reflecting non-barrier disrupted skin [15], in mild AD patients used in this study, or to a limited capacity of POLG to stabilize multi-lamellar structures as acylceramides can [23,32]. Taken together, this clinical study demonstrates the therapeutic value of the POLG nano-emulsion, which can ameliorate the clinical dry skin symptoms associated with mild AD as a consequence of improving the water-holding functions.

The up-regulating effect of the POLG nano-emulsion on the water level in the skin surface that attained a 199.7% improvement in water content (conductance) of the SC at 6 weeks is distinctly prominent in comparison with 175% (conductance) for the CER cream and 157% (conductance) for the HIRU lotion at 4 weeks [26], 169% (conductance) for the CER cream and 150% (conductance) for the urea cream at 6 weeks [31], and 151% (capacitance) for the CER cream and 126% (Capacitance) for the HIRU lotion at 4 weeks [28] in clinical studies of AD dry skin. Thus, it is likely that substances such as CER [17] and POLG [29,30] that have a lamellae-forming ability are better suited to up-regulate the water-holding function than substances such as HIRU and urea that do not have that ability, since a lamellae-forming ability is definitely associated with the capacity to incorporate non-evaporable bound water [22] into lamellar structures which have roles in preventing the evaporation of water molecules even under extremely dry circumstances. 

Significant benefits from the POLG nano-emulsion were noted, specifically in the clinical scoring of scaling, irritating sensation, and itchiness. Those three scorings were significantly alleviated at 3 weeks and at 6 weeks by treatment with the POLG nano-emulsion compared to week 0. The significant improvement in the scaling score is also distinctly supported by the significantly decreased and increased index by Visioscan for roughness (SESC) and smoothness (SESM) at 3/6 and 6 weeks, respectively. This improvement in scaling score also supports the fact that it appears concomitant with the significantly increased corneocyte size, the significantly down-regulated degree of thick abrasions, and the significantly impaired corneocyte lipid envelope. Further, the 199.7% improvement in water content of the SC at 6 weeks strongly suggests that the POLG nano-emulsion regulates the water-holding function in the SC, probably via the increased formation of lamellar structures into which water molecules are incorporated as non-evaporable bound water. The coordinated alleviations of all those factors strongly support the possibility that application of the POLG nano-emulsion to the skin may accelerate lamellae formation between the SC layers to enhance its water-holding function, resulting in a marked amelioration of desquamation, as well as a down-regulation of the turnover rate, which is responsible for regulating corneocyte size [33]. 

During our previous clinical studies on AD dry skin using the CER cream or the HIRU lotion [26,28], the significant amelioration of the clinical scoring of scaling and itchiness was accompanied by a significantly increased water content (conductance), as well as a significantly up-regulated barrier function (TEWL), which had previously suggested that both the up-regulated barrier and water functions are essential requirements for amelioration of scaling and itchiness. However, the improvement in scaling and itchiness elicited by treatment with the POLG nano-emulsion in the present study occurred concomitantly with a significant increase in water content, but was not accompanied by a recovery of the barrier function as expressed by a down-regulated TEWL. Thus, our clinical study using the POLG nano-emulsion strongly suggests that the increased water content plays a more critical role in mitigating scaling and itchiness than the up-regulated barrier function in AD skin. On the other hand, it is likely that the disrupted barrier function in AD skin is mainly attributable to the vulnerability for irritation and allergic reactions by foreign substances such as perspiration and mite antigens, as we have already demonstrated using photo-acoustic spectrometry (PAS) [16]. PAS analysis has revealed that the penetration behavior of hydrophilic materials is a better reflection of TEWL, and the penetration rates of hydrophilic materials are paralleled by the severity of AD in concert with a significant correlation with serum IgE levels in the severe AD group. It should be noted that serum IgE levels represent a hallmark of the atopic diathesis, and about 80% of AD patients have increased levels of serum IgE [34]. It was also suggested that the abnormal barrier functions of AD dry skin may predispose to inflammatory processes evoked by irritants and allergens, especially water-soluble elements such as mite antigens [16].

This study has several limitations: first, we did not compare different vehicles and solvents to deliver the bioactives to the skin, and it is possible that other vehicles might be more or less efficient than the POLG nano-emulsion. Second, we do not have any preclinical data on the AD patients of this study concerning the disrupted water-holding function of their skin. Third, the SC water-holding capacity of the skin is also, in part, regulated by filaggrin in corneocytes, which can vary significantly in different subjects, and the filaggrin genotypes of the AD patients in this study were not assessed. Although clinical evaluations using the 10% POLG nano-emulsion on atopic dry skin should be conducted as a double-blind study with controls, which requires a non-treatment side, we did not set up a control side for the convenience of the enrolled patients with AD. Because this clinical test was carried out during the period from 10/15/2021 (autumn) to 12/17/2021 (winter) in Tokyo, Japan, it is well recognized that climate changes may not ameliorate the atopic dry skin.

In conclusion, taken together, our clinical study suggests, for the first time, that the topical application of the POLG nano-emulsion has a distinct potential to ameliorate atopic dry skin symptoms, especially scaling and itchiness, in the skin of patients with AD by alleviating the disrupted water-holding function as a result of supplying lamellar structures into the SC despite the failure to improve barrier functions. This study also provides an insight into developing an effective skin care system for AD; it seems reasonable to assume that materials with the ability to form lamellar structures, even non-ceramide derivatives, are also suitable to ameliorate atopic skin symptoms in mild AD patients without severely disrupted barrier function. 

## 4. Materials and Methods

### 4.1. Materials

The nano-emulsion containing 10% phytosteryl/octyldodecyl lauroyl glutamate (POLG) was provided by ToutVert Co., Ltd. (Osaka Japan). The chemical structure of POLG is shown in Figure 11, and the components of the POLG nano-emulsion formula are shown in Table 1.

### 4.2. Study Design

This study (TV-2021-10-001) was performed from 15 October 2021, to 17 December 2021, at the Ebisu Skin Research Center, Inforward Co., Ltd., Tokyo, Japan. The nano-emulsion containing 10% POLG was applied twice a day for 6 weeks on the right or left forearm of 25 Japanese patients with AD.

### 4.3. Patients 

Diagnoses of AD were made according to the criteria of Hanifin and Rajka [35]. The severity of AD was evaluated on the basis of the Rajka and Langeland standard with 1–3 grades: 1 = mild; 2 = moderate; 3 = severe [36]. AD patients who visited the Ebisu Skin Research Center, Inforward Co., Ltd. (Tokyo, Japan) were enrolled in the study. In enrolling AD patients with a mild severity, complications with ichthyosis vulgaris and senile xerosis were excluded. Measurements were performed on the uninvolved forearm skin of patients with AD (n = 25; 25 females, aged from 30 to 50 years; mean age, 45 years). In this study, dry skin sites without erythema were chosen as the nonlesional skin for the measurements. 

### 4.4. Compliance with Ethical Standards and Ethical Approval 

This clinical test was registered under the number UMIN000037554. All procedures using human subjects were performed in accordance with the ethical standards of the Institutional and/or National Research Committee, and with the 1964 Helsinki Declaration and its later amendments or comparable ethical standards. The present study adhered to the tenets of the Declaration of Helsinki, and was reviewed and approved by the Japanese Society of Aesthetic Dermatology Examination Ethics Committee (CrrC21-031). Formal informed consent was obtained from each subject before the study. Following screening by a dermatologist, 25 Japanese patients with AD with mild severity attended this evaluation.

### 4.5. Clinical Assessments

A trained dermatologist (KN) performed all clinical assessments. The evaluation parameters consisted of dryness, erythema, papules, and abrasions, and were scored on a 1–5 grading basis (1 = none; 2 = slight; 3 = mild, 4 = moderate; 5 = severe). Sensory attributes, such as stimulus sensations and itchiness, were assessed by querying each subject, and were scored on the same 1–5 grading basis.

Clinical improvement was scored on a 0–3 grading basis (3 = marked improvement, 2 = moderate improvement, 1 = slight improvement, 0 = no improvement) by comprehensively evaluating the clinical scoring. Thus, clinical improvement was evaluated according to the difference in dryness/scaling and itchiness score between before and after treatment as follows: when either score changed by one grade compared with before treatment, a slight improvement (1) was assigned. When both scores changed by one grade compared with before treatment, a moderate improvement (2) was assigned. When either or both scores changed by more than two grades compared with before treatment, a marked improvement (3) was assigned. Thus, it is likely that clinical improvement substantially reflects an efficacy rate for the amelioration of atopic dry skin symptoms.

### 4.6. Evaluation of Skin Microrelief Parameters 

According to previously described methods [37,38], skin microrelief parameters were evaluated on the forearm skin using a VisioScan VC98 (Courage & Khazaka), which is a special high-resolution UV-A light video camera developed specifically to study the skin surface directly, allowing qualitative and quantitative assessments under physiological conditions in scanning images. The gray-level distribution of each image was used to evaluate the following skin parameters: skin roughness (SER), skin scaliness (SESC, the level of dryness of the SC), and number and width of wrinkles (SEW).

### 4.7. Biophysical Measurements 

Conductance, a parameter that reflects the water content of the skin surface, was measured using a SKICON-200EX-USB (Yayoi Co., Ltd., Tokyo, Japan), and is expressed as µS. The measurements were repeated five times within a 2 cm^2^ area on the forearm of each individual, and the average of three values was used after removing the maximum and minimum values. TEWL, a parameter for the cutaneous barrier function of the SC, was measured using a Tewameter TMHex (Courage and Khazaka GmbH, Germany), and is expressed as g/m^2^·h. The measurements were repeated five times within a 2 cm^2^ area on the forearm of each individual, and the average was used. All measurements were performed in a climate-controlled room at 21 °C with an average humidity of 51.62% (range, 47–55%). To minimize regional anatomic differences, the ventral forearm was used as the measuring site whenever possible. 

### 4.8. Collection and Transfer of Corneocytes to Glass Slides

Corneocytes were obtained by the tape-stripping method using cellophane tape (Nichiban, Tokyo, Japan). The tape was cut into a circle with a diameter of 6 mm that was tightly adhered to a glass slide to transfer corneocytes. Each glass slide was then immersed in xylene twice and then in ethanol to remove the cellophane by dissolving the adhesive agent. Corneocytes transferred to glass slides were subjected to the following examinations.

### 4.9. Corneocyte Size and Degree of Thick Abrasions

Measurements of corneocyte size and the degree of thick abrasions were performed according to previously described methods [37]. In brief, each glass slide was immersed in an aqueous solution containing 0.5% brilliant green and 1.0% gentian violet (BG dye solution) for an adequate time (5–10 min); after which, it was rinsed with water. The corneocyte size and the percentage of thick abrasions were analyzed using images taken via microscopy with specialized corneocytometry software (CIEL, Osaka, Japan).

### 4.10. Staining of Lipid Envelopes

Measurements of the staining of lipid envelopes were performed according to previously described methods [37]. In brief, corneocytes on glass slides were hydrated with phosphate buffered saline (PBS–) for 5 min. After blocking with Block Ace (Bio-Rad, Hercules, CA, USA) for 1 h, the lipid envelopes were stained with 3 μg/mL Nile red dye in 75% glycerol, and were then covered with a cover glass. Lipid envelopes were observed as red fluorescence using a fluorescence microscope (Floid Cell Imaging Station). Fluorescence intensities were determined for each fluorescence image using corneocytometry software (CIEL, Osaka, Japan).

### 4.11. Statistical Analyses

Statistical analyses were performed using GraphPad Prism 9 (GraphPad Software, San Diego, CA, USA). For clinical data, we used ANOVA with the Dunn’s multiple comparisons test.

For other data, normality was confirmed by the Shapiro–Wilk test. Data with more than three groups were analyzed by ANOVA, followed by Tukey’s multiple comparison test for parametric data, or by Dunn’s multiple comparison test for nonparametric data. Data with two groups were analyzed by a paired t-test for parametric data, or by the Wilcoxon matched-pairs signed rank test for non-parametric data. *p*-values of 0.05 or less were considered statistically significant.

## Figures and Tables

**Figure 1 ijms-23-13362-f001:**
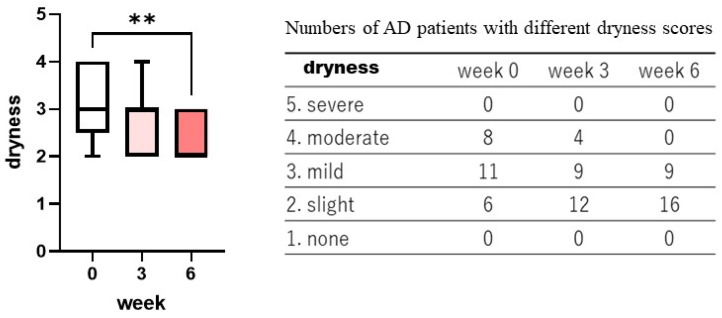
Clinical effects of the POLG nano-emulsion on the dryness of the skin of AD patients. Dryness scoring was performed by a dermatologist (KN) at 0, 3, and 6 weeks on a 1–5 grading basis (1 = none; 2 = slight; 3 = mild, 4 = moderate; 5 = severe). n = 25; **: *p* < 0.01 compared to week 0; all data were analyzed using Dunn’s multiple comparisons test.

**Figure 2 ijms-23-13362-f002:**
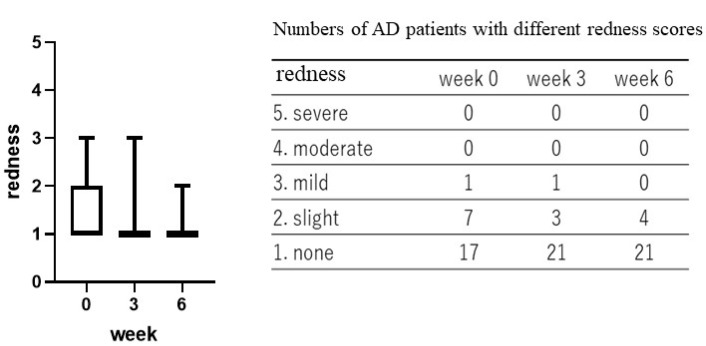
Clinical effects of the POLG nano-emulsion on redness of the skin of AD patients. Redness scoring was performed by a dermatologist (KN) at 0, 3, and 6 weeks on a 1–5 grading basis (1 = none; 2 = slight; 3 = mild, 4 = moderate; 5 = severe). n = 25. All data were analyzed using Dunn’s multiple comparisons test.

**Figure 3 ijms-23-13362-f003:**
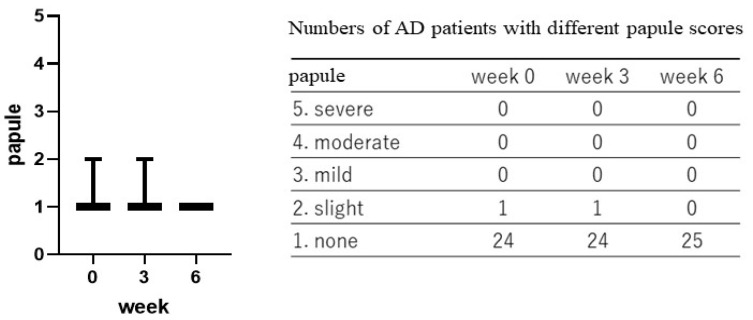
Clinical effects of the POLG nano-emulsion on papules of the skins of AD patients. Papule scoring was performed by a dermatologist (KN) at 0, 3, and 6 weeks on a 1–5 grading basis (1 = none; 2 = slight; 3 = mild, 4 = moderate; 5 = severe). n = 25. All data were analyzed using Dunn’s multiple comparisons test.

**Figure 4 ijms-23-13362-f004:**
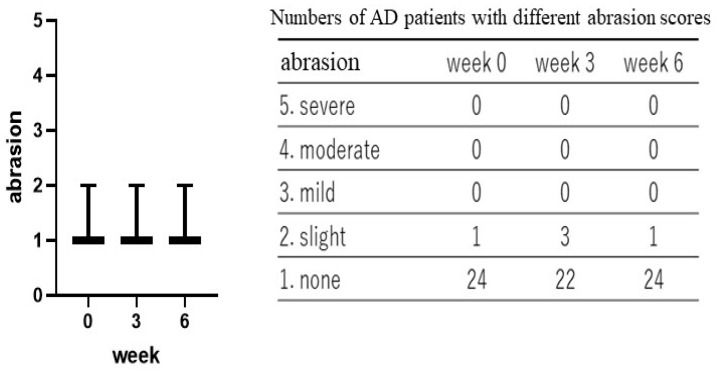
Clinical effects of the POLG nano-emulsion on abrasions of the skins of AD patients. Abrasion scoring was performed by a dermatologist (KN) at 0, 3, and 6 weeks on a 1–5 grading basis (1 = none; 2 = slight; 3 = mild, 4 = moderate; 5 = severe). n = 25. All data were analyzed using Dunn’s multiple comparisons test.

**Figure 5 ijms-23-13362-f005:**
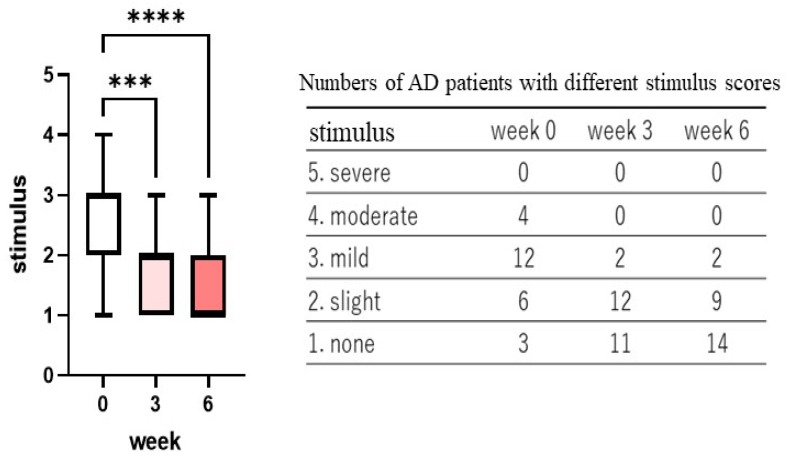
Clinical effects of the POLG nano-emulsion on stimulus sensations of the skin of AD patients. Stimulus sensations were assessed by a dermatologist (KN) after querying each subject at 0, 3, and 6 weeks, and were scored on a 1–5 grading basis (1 = none; 2 = slight; 3 = mild, 4 = moderate; 5 = severe). n = 25; ***: *p* < 0.001, ****: *p* < 0.0001 compared to week 0; all data were analyzed using Dunn’s multiple comparisons test.

**Figure 6 ijms-23-13362-f006:**
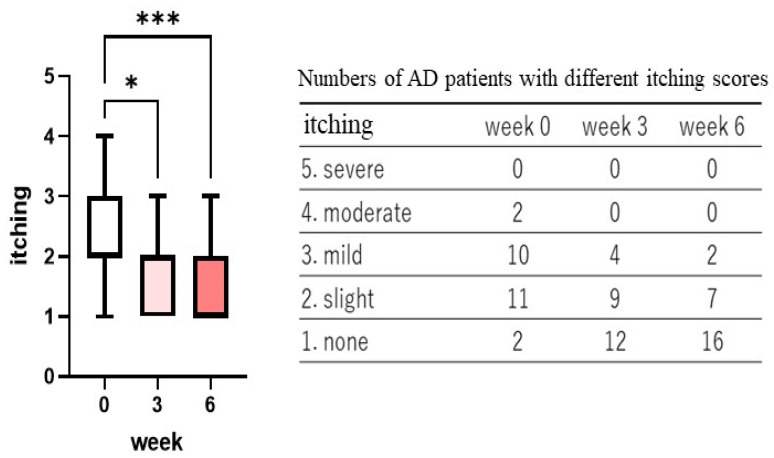
Clinical effects of the POLG nano-emulsion on itchiness of the skin of AD patients. Itchiness sensations were assessed by a dermatologist (KN) after querying each subject at 0, 3, and 6 weeks, and were scored on a 1–5 grading basis (1 = none; 2 = slight; 3 = mild, 4 = moderate; 5 = severe). n = 25; * *p* < 0.05, ***: *p* < 0.001 compared to week 0; all data were analyzed using Dunn’s multiple comparisons test.

**Figure 7 ijms-23-13362-f007:**
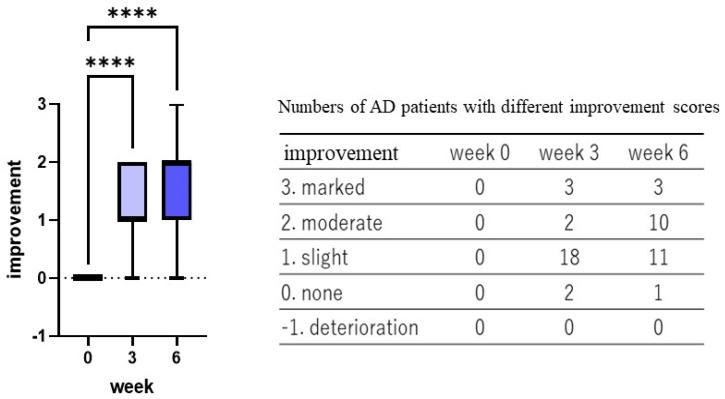
Clinical improvement of the POLG nano-emulsion on AD skin. Clinical improvement was assessed by a dermatologist (KN) at 3 and at 6 weeks after comprehensively evaluating the clinical scoring, n, and was scored on a 0–3 grading basis (3 = marked improvement, 2 = moderate improvement, 1 = slight improvement, 0 = no improvement), as described in the Materials and Methods section. n = 25; ****: *p* < 0.0001 compared to week 0; all data were analyzed using Dunn’s multiple comparisons test.

**Figure 8 ijms-23-13362-f008:**
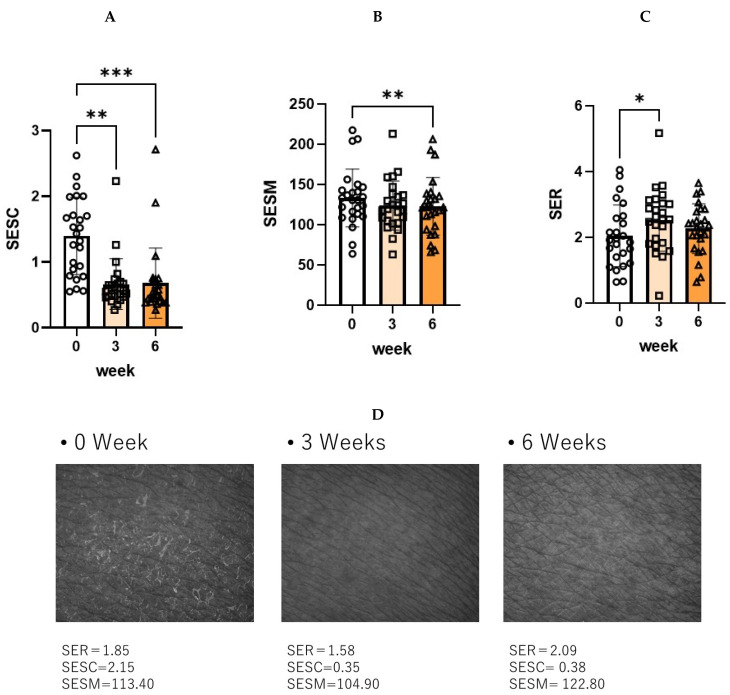
Skin Surface Evaluation by Visioscan. (**A**): Scaliness (SESC) n = 25, (**B**): Smoothness (SESM), n = 25 (**C**): Roughness (SER) n = 25, ***: *p* < 0.001, **: *p* < 0.01, *: *p* < 0.05 compared to week 0, all data were analyzed by Tukey’s multiple comparison test and are expressed as means ± SD. (**D**): Representative photos at 0, 3, and 6 weeks.

**Figure 9 ijms-23-13362-f009:**
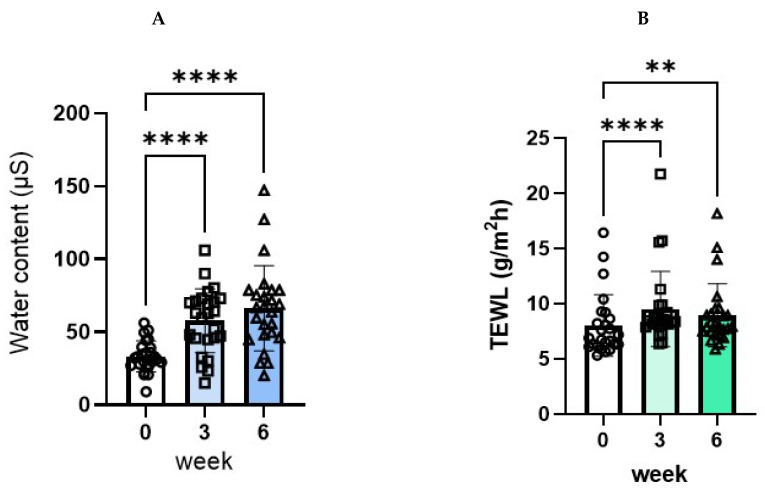
Effects of the POLG nano-emulsion on water content and TEWL. (**A**): Water content: n = 25, all data were analyzed using Tukey’s multiple comparison test and are expressed as means ± SD. (**B**): Barrier function: n = 25, all data were analyzed using Dunn’s multiple comparison test and are expressed as means ± SD. **: *p* < 0.01, ****: *p* < 0.0001 compared to week 0.

**Figure 10 ijms-23-13362-f010:**
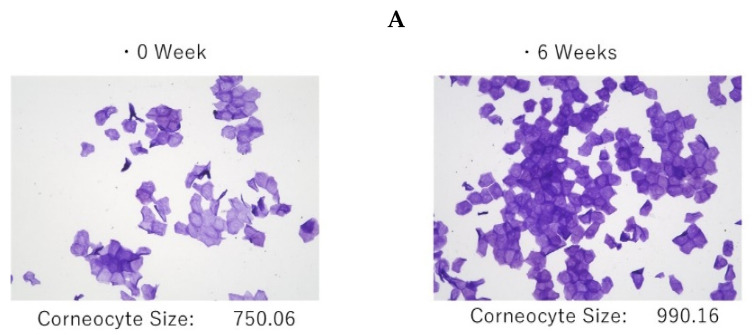
Effects on corneocyte size, degree of thick abrasions, and Nile red dye staining in the tape-stripped SC. (**A**): Corneocyte size, n = 25; ***: *p* < 0.001, compared to week 0; data were analyzed using the Wilcoxon signed-rank sum test and are expressed as means ± SD. (**B**): Thick abrasions, n = 25; ****: *p* < 0.0001, compared to week 0; data were analyzed using the paired t test and are expressed as means ± SD. (**C**): Nile red dye staining, n = 25; *: *p* < 0.05, compared to week 0; data were analyzed using the paired t test and are expressed as means ± SD.

**Figure 11 ijms-23-13362-f011:**
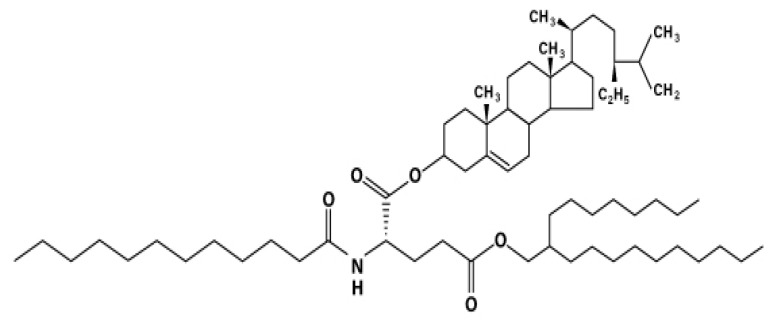
Chemical structure of phytosteryl/octyldodecyl lauroyl glutamate (POLG).

**Table 1 ijms-23-13362-t001:** Components of the POLG nano-emulsion formula.

Water
Phytosteryl/Octyldodecyl Lauroyl Glutamate (POLG); 10.0%
Glycerin
Butylene Glycol
Dipropylene Glycol
Pentylene Glycol
Hydrogenated Lecithin
Lactobacillus/Wasabia Japonica Root Ferment Extract
Polyglyceryl-10 Laurate
Sodium Pyrrolidone Carboxylic Acid
Ceramide 3
Pyrrolidone Carboxylic Acid
Arginine
Aspartic Acid
Glycine
Alanine
Serine
Valine
Isoleucine
Threonine
Proline
Histidine
Phenylalanine
Saccharide Isomerate
Rosmarinus Officinalis (Rosemary) Leaf Extract
Polyquaternium-51
Lactobacillus Ferment
Phenoxyethanol
Sodium Lactate

## Data Availability

This clinical test was registered under the number UMIN000037554.

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
