# Peer review of "A Nano-Emulsion Containing Ceramide-like Lipo-Amino Acid Cholesteryl Derivatives Improves Skin Symptoms in Patients with Atopic Dermatitis by Ameliorating the Water-Holding Function"

_ijms, 2022, doi:10.3390/ijms232113362_

Round 1

Reviewer 1 Report

 Takada et al. performed a clinical study that evaluated the efficacy of topically delivered nano-emulsion-containing phytosterol/octyl dodecyl lauroyl glutamate (POLG) on individuals with atopic dermatitis (AD).

 POLG nano-emulsion improved AD clinical symptoms, skin surface smoothness, and skin surface water content (assessed by size, coherence, and lipophilicity of corneocytes).

 The manuscript clearly presented data suggesting topical POLG nano-emulsions' efficacy for sensitive skin individuals.  

 This reviewer raises some issues.

Major

1. This study does not have control arms. Efficient drug delivery into the stratum corneum (SC) depends on the nature of the vehicle/solvent used. Despite many speculative statements in the discussion section, the authors did not describe the study design limitation. At least they should include limitation statements in the discussion section.

2. Along the same line, the authors should provide preclinical data (either in vitro or in vivo) supporting the reinforcement function on the SC that they claimed, if available. If not, this should also be incorporated into the limitation statements.

3. SC water holding capacity is also attributable to humectants stored in corneocytes, primarily affected by filaggrin genotypes. The authors should provide the genetic component of the AD individuals studied if data are accessible.

Minor

2. What do CER and HIRU creams stand for? The authors should spell out the abbreviations. 

Author Response

Responses to Reviewer 1

 Comments and Suggestions for Authors

 Takada et al. performed a clinical study that evaluated the efficacy of topically delivered nano-emulsion-containing phytosterol/octyl dodecyl lauroyl glutamate (POLG) on individuals with atopic dermatitis (AD).

 POLG nano-emulsion improved AD clinical symptoms, skin surface smoothness, and skin surface water content (assessed by size, coherence, and lipophilicity of corneocytes).

 The manuscript clearly presented data suggesting topical POLG nano-emulsions' efficacy for sensitive skin individuals. 

 This reviewer raises some issues.

Major

  1. This study does not have control arms. Efficient drug delivery into the stratum corneum (SC) depends on the nature of the vehicle/solvent used. Despite many speculative statements in the discussion section, the authors did not describe the study design limitation. At least they should include limitation statements in the discussion section.

Response: Considering this comment, we have added this limitation to the statement in the revised Discussion section, as marked by red color just before the conclusion paragraph.

----------------------------------------------------

  1. Along the same line, the authors should provide preclinical data (either in vitro or in vivo) supporting the reinforcement function on the SC that they claimed, if available. If not, this should also be incorporated into the limitation statements.

Response: Unfortunately, we have no preclinical data (either in vitro or in vivo) for these patients and have added this limitation to the statement in the Discussion section, as marked by red color just before the conclusion paragraph.

  1. SC water holding capacity is also attributable to humectants stored in corneocytes, primarily affected by filaggrin genotypes. The authors should provide the genetic component of the AD individuals studied if data are accessible.

Response: Considering this comment, because we have no data about the genetic component of the AD patients enrolled in this study, we have noted this in the limitations statement in the Discussion section, as marked by red color just before the conclusion paragraph.

Minor

  1. What do CER and HIRU creams stand for? The authors should spell out the abbreviations.

Response: We have clarified those abbreviations as marked by red color in the revised Discussion section.

----------------------------------------------------------------------

Reviewer 2 Report

The manuscript is well written. 

Author Response

Response to Reviewer 2:

Comments and Suggestions for Authors

The manuscript is well written.

Response: Thank you very much for your positive comment.

------------------------------------------------------------------------------

Reviewer 3 Report

Dear Authors

Unfortunately, the current quality of this manuscript including figures, unkind descriptions of each figure, and references is not enough and is unsuitable to publish in IJMS considering the recent quality of IJMS. 

Here are the major comments. 

1. Why authors choose 10 % phytosteryl/octyldodecyl lauroyl glutamate (POLG)? What's the basis of this? Authors should include this relevant data after major revision. 

2. All figures should contain sufficient backgrounds and describe each content in detail. Only 4-6 lines are not so enough. There are no references except for Figure 4. 

3. Please, describe the definition of clinical improvement in Figure. 2, but not only a method. References should be needed.  

4. Figures 1 and 2 need high-quality images. 

5. English editing service will be helpful for clarifying the manuscript. 

Best Regards 

Author Response

Responses to Reviewer 3:

Comments and Suggestions for Authors

Dear Authors

Unfortunately, the current quality of this manuscript including figures, unkind descriptions of each figure, and references is not enough and is unsuitable to publish in IJMS considering the recent quality of IJMS.

Response: Thank you for reviewing our manuscript carefully and for providing such helpful comments and suggestions to clarify and improve it. In response to this comment, we have modified all the original Figures (1 – 6) to clarify them and have added text to the legend of each Figure to detail how the results were obtained and analyzed. Those original Figures are now Figures 1 – 11 in the revised manuscript.

-----------------------------------------------------------

Here are the major comments.

  1. Why authors choose 10 % phytosteryl/octyldodecyl lauroyl glutamate (POLG)? What's the basis of this? Authors should include this relevant data after major revision.

Response: In response to this comment, in order to clarify the basis of using 10% phytosteryl/octyldodecyl lauroyl glutamate (POLG), we have added a detailed explanation about why we chose that chemical in the revised Introduction section, as marked by red color.

----------------------------------------------------------------------------------

  1. All figures should contain sufficient backgrounds and describe each content in detail. Only 4-6 lines are not so enough. There are no references except for Figure 4.

Response: In response to this comment, we have modified original Figures 1 - 6 (now Figures 1 – 11 in the revised manuscript) and have added detailed text to the legend of each Figure, as marked by red color.

-------------------------------------------------------------------

  1. Please, describe the definition of clinical improvement in Figure. 2, but not only a method. References should be needed.

Response: Considering this comment, we have added the method used and the definitions of clinical improvement in detail, as marked by red color in Clinical Evaluation section of the Materials and Methods and briefly in the Clinical Improvement section of the Results.

---------------------------------------------------------------------------------------

  1. Figures 1 and 2 need high-quality images.

Response: As requested, we have changed original Figures 1 and 2 with high-quality images (now Figures 1 – 7 in the revised manuscript).

--------------------------------------------------------------------------

  1. English editing service will be helpful for clarifying the manuscript.

Response: Our revised manuscript has been edited by an English scientific manuscript editing service, DASS Manuscript. Please find an attached file of its certification statement.

--------------------------------------------------------------------------------

Round 2

Reviewer 1 Report

Authors have addressed the issues raised by the reviewer appropriately.

Author Response

Reponse;

Thank you very much for a positive comment.

Reviewer 3 Report

Dear authors

This revised manuscript is much better.

Here are minor comments.  

1. There is no explanation or description about additional contents of Figures in this revised manuscript. Especially, dryness, erythema, papules and abrasions of atopic dermatis in introduction.

2. There are small number of references. Please, add referecences more. 

Best Regards,  

Author Response

Reviewer 3

This revised manuscript is much better.

Here are minor comments.  

  1. There is no explanation or description about additional contents of Figures in this revised manuscript. Especially, dryness, erythema, papules and abrasions of atopic dermatitis in introduction.

Response: Considering the reviewer’s comments, we have added explanation or description about dryness, erythema, papules and abrasions of atopic dermatitis in Introduction section as marked by red color.

---------------------------------------------------------------------------------------------------------------------

  1. There are small number of references.  Please, add references more.

Response: Considering the reviewer’s comments, we have added 8 new references through the re-revised manuscript and in References section as marked by red color.